# OpenReview forum: "Leveraging System-Prompt Attention to Counteract Novel Jailbreak Attacks"
_ICLR.cc/2025/Conference — Submitted to ICLR 2025_

### Official Review · Reviewer_2MKg · 2024-10-27

**Soundness:** 1
**Presentation:** 2
**Contribution:** 1
**Rating:** 3
**Confidence:** 4

**Summary:**

This paper introduces ALMAS, a multi-agent framework for generating novel jailbreak prompts and then proposes AttentionDefense, a novel approach that uses system prompt attention patterns from Small Language Models (SLMs) to detect both known and novel jailbreak attacks. Experiments show that SLM-based AttentionDefense has equivalent or better jailbreak detection performance as compared to text embedding based classifiers and GPT-4 zero-shot detectors.

**Strengths:**

- The paper demonstrates that system-prompt attention from Small Language Models (SLMs) can be used to detect jailbreak prompts providing a novel explainable and cheaper defense approach called AttentionDefense.
- The paper points out that system prompt attention may perform better than semantic embedding classification methods because it measures the LM's response to attempts on overriding safety mechanisms.

**Weaknesses:**

- The novelty and necessity of ALMAS is not sufficiently justified in the context of existing jailbreak attack methods. Prior work such as AutoDan [1], GPTFuzzer [2], and PAIR [3] have proposed automatic methods generating jailbreak prompts through template-based mutation and optimization. The paper does not adequately demonstrate how ALMAS achieve the state-of-the-art performance in jailbreak attack.
- The validation of ALMAS's effectiveness is insufficient. Without comprehensive evaluation against existing state-of-the-art jailbreak attack methods, the robustness of AttentionDefense against advanced attacks remains uncertain.
- The experimental evaluation would benefit from more comprehensive comparisons with other defense approaches. While the paper demonstrates advantages over embedding-based methods and GPT-4 detection, it lacks comparison with recent specialized defense mechanisms such as Safedecoding [4].


[1] Liu, Xiaogeng, et al. "Autodan: Generating stealthy jailbreak prompts on aligned large language models." arXiv preprint arXiv:2310.04451 (2023).

[2] Yu, Jiahao, et al. "Gptfuzzer: Red teaming large language models with auto-generated jailbreak prompts." arXiv preprint arXiv:2309.10253 (2023).

[3] Chao, Patrick, et al. "Jailbreaking black box large language models in twenty queries." arXiv preprint arXiv:2310.08419 (2023).

[4] Xu, Zhangchen, et al. "Safedecoding: Defending against jailbreak attacks via safety-aware decoding." arXiv preprint arXiv:2402.08983 (2024).

**Questions:**

-  How does ALMAS compare with other state-of-the-art jailbreak attack methods like AutoDAN, GPTFuzzer, and PAIR in terms of attack success rate and efficiency?
- Could the authors provide  experiments between AttentionDefense and recent defense mechanisms like Safedecoding?

---

### Official Review · Reviewer_1mLw · 2024-11-03

**Soundness:** 2
**Presentation:** 1
**Contribution:** 2
**Rating:** 3
**Confidence:** 4

**Summary:**

The paper presents an innovative end-to-end framework designed to counteract jailbreak attacks on Language Models (LMs). Central to this framework is ALMAS (Attack using LLM-based Multi-Agent Systems), a novel mechanism for generating adversarial attacks, and AttentionDefense, a strategy for defending against them. The study underscores the limitations of existing jailbreak detection methods, noting issues such as limited generalizability and poor scalability. By harnessing attention weights from Small Language Models (SLMs), the proposed AttentionDefense method aims to offer an explainable, computationally efficient, and effective approach for detecting both known and novel adversarial prompts.

**Strengths:**

1. AttentionDefense is the first mitigation strategy to leverage system prompt attention for detecting adversarial attacks and the first open-box jailbreak detection classifier utilizing an SLM.

2. Unlike black-box classifiers, AttentionDefense provides explanations for its decisions by analyzing attention weights, enhancing trust and interoperability in its outputs.

3. The use of SLMs makes this method computationally more efficient than relying on larger LLMs, such as GPT-4, while maintaining comparable detection performance.

4. The paper shows that AttentionDefense is effective in identifying both established and novel jailbreaks, outperforming embedding-based classifiers and achieving performance on par with LLM-based detectors.

**Weaknesses:**

1. The paper’s content organization is unclear, leading to a sense of disarray. While the mechanisms of ALMAS and AttentionDefense are not inherently complex, their descriptions in the paper are overly verbose and lack focus. The authors should restructure the content for a smoother logical flow and provide concise overviews in relevant sections to help readers grasp the key points more quickly.

2. The experimental section lacks a clear description of the setup, especially regarding the datasets used for testing. The paper should explicitly list all datasets used in training and evaluation, detailing their sources and contents. It should identify known datasets, the types of jailbreak methods they include, and whether recent attack techniques are covered to ensure a comprehensive evaluation. For novel jailbreak datasets, the authors should explain how these differ in distribution from known datasets and why they are considered novel.

3. The study does not include a comprehensive evaluation of the jailbreak prompts generated by ALMAS. The authors should report the attack success rates of ALMAS prompts across different LLMs to validate their effectiveness and novelty. A comparison between ALMAS-generated prompts and existing jailbreak prompts would help assess their innovation and difficulty, ensuring that ALMAS prompts genuinely possess novel characteristics.

4. The attack generation method used by ALMAS appears to share similarities with the ICAG method [1], which also involves multiple agents leveraging existing jailbreak prompts to generate new ones. The authors should discuss the similarities and differences between these methods in detail and highlight any innovative aspects or superior performance of ALMAS.

5. Figure 3 has font sizes that are too small to read, impairing comprehension. The authors should consider redesigning or enlarging these figures to enhance readability and improve the effectiveness of information delivery.

6. Although the paper mentions the shortcomings of current defense methods, it does not provide a baseline comparison with any existing techniques. The authors should include comparative experiments with several representative defense methods to demonstrate the advantages and effectiveness of AttentionDefense.

7. The name “AttentionDefense” implies a defense mechanism, but it functions more as a detection system. The authors should consider a more appropriate name, such as “AttentionDetection,” to avoid confusion. Additionally, the detection method should be evaluated on generic benign queries (e.g., MMLU) to test for false positives. Given that filter-based defense mechanisms can have high rejection rates for “seemingly harmful” inputs, it would be helpful to test on datasets like PHTest [2] to verify robustness and stability under benign input conditions.

[1] Zhou Y, Han Y, Zhuang H, et al. Defending jailbreak prompts via in-context adversarial game[J]. arXiv preprint arXiv:2402.13148, 2024.

[2] An B, Zhu S, Zhang R, et al. Automatic pseudo-harmful prompt generation for evaluating false refusals in large language models[J]. arXiv preprint arXiv:2409.00598, 2024.

**Questions:**

The same as the weaknesses.

---

### Official Review · Reviewer_a8rp · 2024-11-04

**Soundness:** 1
**Presentation:** 1
**Contribution:** 2
**Rating:** 1
**Confidence:** 3

**Summary:**

The paper presents a framework for detecting jailbreak attacks on Language Models (LMs) through a method called AttentionDefense. This approach utilizes system prompt attention from Small Language Models (SLMs) to identify adversarial prompts effectively. The authors propose ALMAS, a self-learning multi-agent system that generates novel attack patterns to evaluate the robustness of the AttentionDefense framework. The proposed method aims to enhance explainability, scalability, and generalizability of jailbreak detection strategies.

**Strengths:**

1. The use of system prompt attention for detection is a novel contribution, addressing the limitations of existing methods that rely heavily on embedding-based classifiers.
2. AttentionDefense is designed to be less computationally intensive than traditional LLM detection methods, making it more practical for real-world applications.

**Weaknesses:**

1. It is recommended to recheck for grammatical errors, such as in L17 with “…while work…”; also, Figure 3 can be optimized by summarizing the text within the image, which would clarify the role of each agent more clearly.
2. Check for potential writing issues that could hinder understanding, such as in L447-448, where the caption for Figure 5 offers two different interpretations for “any points below,” and both coordinates in Figure 5(c) are labeled as “Novel F1 Score.”
3. The structure of the paper could be better organized. For instance, the data and analysis presented in Section 5 (Results) are quite limited, while Sections 2-4 contain a great deal of detail. Additionally, the layout of Figures 1-3 is not sufficiently compact. There is extensive groundwork and speculation laid out earlier, yet the data analysis and discussion in the Results section are relatively weak. For example, the conclusion in Section 5.2 stating “ATTENTION GENERALIZES BETTER THAN EMBEDDINGS” is one of the key contributions of this paper (mentioned in the introduction), but the discussion surrounding it is quite weak (L406-L414).
4. It is advisable to provide more discussion on the four key research contributions of this paper mentioned in the Introduction, particularly points 2-4. Currently, there is limited evidence and discussion supporting these conclusions. I recommend including ablation studies on LMs to demonstrate the generalizability of these conclusions, as well as conducting a more thorough analysis of the existing data to clarify the logic.

**Questions:**

1. In many LM usage scenarios, the system prompt is either empty or does not contain safety content, such as “You are a chat assistant.” How effective is AttentionDefense in these cases?
2. Can AttentionDefense be used for LLMs?
3. How reproducible are the experiments?

---

### Meta-Review · Area_Chair_3ELD · 2024-12-13

**Metareview:**

The paper builds an LLM-based agent system to generate jailbreak attack examples and a defense method AttentionDefense based on system prompt attention from Small Language Models (SLMs) to identify adversarial prompts. While using system prompt attention as a detection and defensive mechanism is interesting and novel, the paper suffers a lot in writing, organization, and experiments. All reviewers unanimously think the paper needs a careful revision on the organization, typos, figure font sizes, and probably a rewrite of the methodology part. Furthermore, the experiments are quite limited and there needs more experiments to verify the proposed defense's effectiveness. Several attempts could be made such as comparing the AttentionDefense with other defense baselines and including more jailbreak attacks.

**Additional Comments On Reviewer Discussion:**

The author didn't provide any rebuttal so there is no author-reviewer discussion.

---

### Decision · Program_Chairs · 2025-01-22

Reject